# Areas of Crush Nuclear Streaming Should Be Included as Tumor Content in the Era of Molecular Diagnostics

**DOI:** 10.3390/cancers15061910

**Published:** 2023-03-22

**Authors:** Yuri Noda, Ryosuke Yamaka, Naho Atsumi, Koichiro Higasa, Koji Tsuta

**Affiliations:** 1Department of Pathology, Kansai Medical University, 2-5-1 Shin-machi, Hirakata 573-1010, Osaka, Japan; 2Department of Pathology and Laboratory Medicine, Kansai Medical University Hospital, 2-3-1 Shin-machi, Hirakata 573-1191, Osaka, Japan; 3Department of Genome Analysis, Institute of Biomedical Science, Kansai Medical University, 2-5-1 Shin-machi, Hirakata 573-1010, Osaka, Japan

**Keywords:** DNA integrity number, percentage of fragments with >200 nucleotides, formalin-fixed paraffin-embedded tissue, next-generation sequencing, RNA integrity number

## Abstract

**Simple Summary:**

To expand the pool of samples available for genetic analysis, the quality and utility of DNA and RNA extracted from degenerated tumor tissues were examined. The DNA obtained from the nuclear streaming samples was preserved, enabling the identification of reliable variants using next-generation sequencing (NGS) analysis. The NGS metrics and variant allele frequency of the nuclear streaming samples did not differ from those of the control. This suggests that non-tumor cells exhibiting nuclear streaming could be used for genetical analysis and therefore, should be considered as tumor content to avoid misinterpreting the variant allele frequency.

**Abstract:**

Degenerated tissues are frequently observed in malignant tumors, but are not analyzed. We investigated whether nuclear streaming and necrosis samples could be used for genetic analysis to expand the sample pool. A total of 81 samples were extracted from small cell carcinoma and lymphoma FFPE tissue blocks and classified into three histological cohorts: 33 materials with well-preserved tumor morphology, 31 nuclear streaming samples, and 17 necrosis samples. DNA and RNA integrity numbers, percentage of RNA fragments with >200 nucleotides, and next-generation sequencing quality metrics were compared among the cohorts. DNA quality did not significantly differ between nuclear streaming materials and materials with well-preserved morphology, whereas that of the necrosis samples was inferior. RNA quality decreased in the following order: materials with well-preserved morphology > nuclear streaming > necrosis. The sequencing metrics did not differ significantly between the nuclear streaming samples and materials with well-preserved morphology, and reliable variants were detected. The necrosis samples extracted from resections exhibited sequencing failure and showed significantly fewer on-target aligned reads and variants. However, variant allele frequency did not differ among the cohorts. We revelated that DNA in nuclear streaming samples, especially within biopsies, could be used for genetic analysis. Moreover, degenerated non-tumor cells should be counted when evaluating tumor content to avoid misinterpreting the variant allele frequency.

## 1. Introduction

Next-generation sequencing (NGS) to detect molecular alterations has emerged as a precise approach for assessing malignant tumors [1,2,3,4,5,6,7,8]. Although surgical resection is the most effective therapy for early-stage solid malignancies, some advanced cases are inoperable. In such cases, histological and molecular diagnoses are conducted using small amounts of materials, such as biopsies and/or cytological specimens [7]. Although liquid components of blood are available for NGS analysis, tissue and cell materials are still required, as they allow for larger numbers of panels and provide more accurate analytical results than liquid materials [7].

The amount of DNA required for NGS is 50–300 ng [4,7], depending on the platform and target enrichment method. Larger gene panels require a greater amount of materials, and low DNA input negatively affects the sensitivity of variant detection [1,2,3,4,5,6,7]. However, DNA yield depends on the size and cellularity of the selected tumor [7,8,9,10]. The expected average DNA yield from a single nucleated cell is 6 pg. Therefore, to obtain 10 ng DNA, approximately 2000 whole cells are required [4]. Sample quantity can be increased when tumor areas with degradation, such as areas of necrosis or crushed nuclear streaming, which have previously been considered difficult to analyze via NGS [7,8,9,10], are considered suitable for molecular diagnosis.

The recommended histological selection criteria for target sequencing (TS) include at least 20% tumor content without degenerative features; accordingly, areas with tumor necrosis and areas abundant in wild-type non-tumor cells (e.g., inflammatory cells) are excluded [7,8,9,10]. Increasing tumor content and DNA yield would increase the NGS success rate [11]; however, the quality of DNA and RNA in tumor tissues exhibiting degenerative changes has not been examined. In fact, necrotic and crushed samples encountered in routine pathology work are excluded from genetic analysis, despite tumor samples derived from patients with malignant tumors, who specifically require genetic treatment, frequently exhibiting such histological changes. Determining whether necrosis and crushed nuclear streaming samples exhibit genomic integrity could facilitate the improvement of tumor content estimation.

In this study, we examined the quality of nucleic acids extracted from histologically degenerated tissues, such as samples with necrosis and nuclear streaming, and performed targeted NGS and evaluated sequencing metrics to confirm whether these samples could be used in genetic therapy. We evaluated their DNA integrity number (DIN), RNA integrity number (RIN), and percentage of RNA fragments with >200 nucleotides (DV_200_), which are commonly used indices to assess nucleic acid quality. For these evaluations, we used materials from small cell lung carcinoma (SC) and diffuse large B-cell lymphoma (DLBCL), in which morphological nuclear degeneration is frequently observed. Our findings can potentially help in increasing the number of samples available for genetic analysis, thereby improving diagnosis and treatment outcomes.

## 2. Materials and Methods

### 2.1. Patient Eligibility and Sample Preparation

We classified 17 SC and 28 DLBCL samples collected at the Kansai Medical University Hospital (Osaka, Japan) between 2017 and 2020 into three histological areas: material with a well-preserved tumor morphological component (solid tumor component with a clear nuclear shape and recognizable outline; Figure 1A), material with nuclear streaming components (component showing linear or lumpy basophilic nuclei, with no clear cell to cell border, no clear outline of individual nuclei, no cell membrane, and no eosinophilic cytoplasm, usually considered to be histological “artifacts”; Figure 1B), and material containing necrosis components (solid tumor component with an eosinophilic morphology, lacking basophilic nuclei, a clear outline of the cell membrane, and cell adhesion; Figure 1C). The three histological components were extracted from the 45 cases that were included in the study. Histological representative areas with >10% wild-type non-tumor cells and <90% tumor cells were excluded from this study. If all three components were unavailable in the same case, or the amount of the components present in that case was low, only the available components were included in the analysis. The final cohort comprised 33 samples without degeneration (8 SC and 25 DLBCL), including 31 with nuclear streaming (13 SC and 18 DLBCL) and 17 with necrosis (3 SC and 14 DLBCL).

Moreover, we examine whether the sample volume under the formalin fixed process used in the present study influences the quality of nucleic acid. Considering that the fixation time in the present study ranged from 12–24 h, the penetration rate of 10% neutral-buffered formalin was 1 mm/h [7], and the SC and diffuse large B-cell lymphoma (DLBCL) samples were divided into three groups, according to the sample size, according to diameter and height: small (≤12 × 12 mm), large (≥24 × 24 mm), and medium (with dimensions other than those of the small and large groups).

This study was approved by the Institutional Review Board of the Kansai Medical University Hospital (approval no. 2020271). Informed consent was obtained from all patients, and the opt-out approach was applied owing to the retrospective design of the study, with no new risk to the participants. Information regarding this study, such as the inclusion criteria and the opportunity to opt-out, was provided on the hospital’s website. The inclusion and exclusion criteria, sample selection, and sample preparation are described in Appendix A.

### 2.2. Assessment of DNA and RNA Quality and Variant Detection Using NGS

DNA and RNA were isolated from each sample, and their purities were measured; their quality was based on RIN, DIN, and DV_200_. DNA was isolated using a QIAamp DNA formalin-fixed, paraffin-embedded (FFPE) Tissue Kit (Qiagen, Hilden, Germany). RNA was isolated using an miRNeasy FFPE Tissue Kit (Qiagen). The purity of the nucleic acid samples was assessed based on their spectrophotometric absorbance at 260–280 nm (A260/A280) using a NanoDrop 3300 spectrophotometer (Thermo Fisher Scientific, Waltham, MA, USA). RIN, DIN, and DV_200_ indices and the concentrations of the extracted DNA and RNA samples were quantified using TapeStation 4150 (2015, Agilent Technologies, Santa Clara, CA, USA) based on the genomic DNA (gDNA) and RNA ScreenTape assays. Owing to instrument performance, RIN, DV_200_, and DIN can provide quality measurements in samples with RNA concentrations >2.0 ng/μL and gDNA concentrations >3.0 ng/μL. Statistical analysis was performed on cases with evaluable RIN, DV_200_, and DIN. Subsequently, to asses DNA quality and detect variants, NGS was conducted using an MiSeq sequencing platform. The AmpliSeq for Illumina Cancer HotSpot Panel v2 (50 target genes, 207 amplicons; Illumina Inc., San Diego, CA, USA) protocol was applied (Appendix A).

### 2.3. NGS Data Analysis and Validation of Observed Variants

NGS data analysis is summarized in Appendix A. Recommended total read depth corresponding to variant allele frequency (VAF) was calculated using https://github.com/mvasinek/olgen-coverage-limit, accessed on 28 January 2022 [12]. When considering clinical reliability, samples with an amplicon mean coverage of <500× were considered non-informative for detecting somatic mutations with a frequency of 5% [8,13].

### 2.4. Statistical Analysis

Because DIN, RIN, and DV_200_ did not differ between DLBCL and SC samples (*p* > 0.05, Appendix A), these two tumors were analyzed together. Differences in DIN, RIN, and DV_200_ were assessed using Welch’s *t*-test, followed by the F test. The percentage of variants that passed the filter with a quality of 100 was calculated as:(1)$$\frac{number\;of\;variants\;that\;passed\;the\;filter\;with\;a\;quality\;of\;100}{number\;of\;all\;detected\;variants}$$

The percentage of variants that did not pass the filter or did not show a quality of 100 was calculated as:(2)$$\frac{number\;of\;variants\;that\;did\;not\;pass\;the\;filter\;+\;number\;of\;reads\;which\;did\;not\;have\;a\;quality\;of\;100}{number\;of\;all\;detected\;variants}$$

Chi-square (*χ^2^*) analysis and Tukey–Kramer or Gomes–Howell tests were used to assess the relationships between materials without degeneration, those with nuclear streaming, and those with necrosis, as well as the differences based on the sample volume (small, medium, and large). Spearman’s correlation was used to assess the relationship between the percentage of on-target aligned reads and DIN. Statistical analysis was performed using SPSS (v20.0; IBM Corp., Armonk, NY, USA), with significance set at *p* < 0.05.

## 3. Results

### 3.1. DIN, RIN, and DV_200_ of Materials without Degeneration, with Nuclear Streaming, and with Necrosis

#### 3.1.1. DIN

DIN was available from 71 samples (range = 1.6–5.5, mean = 2.8 ± 0.12, Appendix A). DIN was significantly lower in necrosis samples than in nuclear streaming samples (*p* < 0.05, Table 1, and Figure 2A). Although no significant difference in DIN was observed between necrosis samples and samples with well-preserved morphology, necrosis samples showed the lowest average DIN among cohorts (average DIN: materials without degeneration 2.78 ± 0.69; nuclear streaming 3.16 ± 1.12; necrosis 2.29 ± 0.53).

#### 3.1.2. RIN

RIN was available from 72 samples (range = 1.1–3.7, mean = 1.87 ± 0.06, Appendix A). Nuclear streaming and necrosis samples showed a lower RIN than materials with well-preserved morphology (vs. nuclear streaming, *p* = 0.05; vs. necrosis, *p =* 0.026; Table 1 and Figure 2B). The average RIN decreased in the following order: materials with well-preserved morphology (2.05 ± 0.60) > nuclear streaming samples (1.76 ± 0.32) > necrosis samples (1.68 ± 0.28).

#### 3.1.3. DV_200_

DV_200_ was available from 79 samples (range = 34.89–90.77, mean = 62.7 ± 1.64, Appendix A). The average DV_200_ value of the DLBCL and SC samples decreased in the following order: materials with well-preserved morphology (68.3 ± 16.4) > nuclear streaming samples (59.6 ± 13.7) > necrosis samples (57.5 ± 8.16; Table 1 and Figure 2C). Nuclear streaming and necrosis samples showed a significantly lower DV_200_ than materials with well-preserved morphology (both *p* < 0.05).

### 3.2. Results of NGS via AmpliSeq for Illumina Cancer HotSpot Panel v2

The average A260/A280 ratio in the three histological cohorts was within the reference range (1.8–2.0), with no significant difference between the cohorts (all *p >* 0.05, Appendix A). RIN values indicated that the samples were too degraded for NGS analysis; RIN > 8 is regarded as usable [7]. Therefore, TS was performed with AmpliSeq, which can be achieved using low-quality DNA.

TS was performed by selecting cases with high and low DIN values. Considering that DNA degradation during storage can affect the TS results [7], TS was also performed using two SC samples obtained in 2015. The NGS results, metrics, and data of the detected variants are summarized in Table 2 and Table 3, and the detailed data on the FASTQ file, and the detected variants are presented in Appendix A.

A total of 25 samples (10 materials with well-preserved morphology, 9 with nuclear streaming, and 6 with necrosis) were obtained from 10 cases and evaluated using NGS. Both samples obtained in 2015 showed an amplicon coverage >500×, suggesting that the storage time did not greatly affect nucleic acid quality. However, amplicons with a coverage <500× were detected in two necrosis samples (SC2018-2, amplicon mean coverage 29.2×; DL2018-19, 82.6×; Figure 3A,B).

Statistical analyses of the 25 samples revealed significantly lower percentages of on-target aligned reads for necrosis samples than for materials with well-preserved morphology and nuclear streaming samples (materials with well-preserved morphology, 95.7 ± 3.02; with nuclear streaming, 94.6 ± 6.44; and with necrosis, 54.1 ± 40.6; all *p* < 0.05, Table 4 and Figure 3C). No significant differences in other NGS metrics were observed among materials with well-preserved morphology, those with nuclear streaming, or those showing necrosis (all *p* > 0.05).

The number of variants detected is shown in Table 3. All 25 samples exhibited the recommended read depth corresponding to VAF. Informative variants were detected in 23 cases, including nuclear streaming and necrosis samples; these included variants in *KDR*, *TP53*, *KIT*, *PTEN*, *KRAS*, *ERBB4*, *MET*, *STK11* in SCs, and *TP53*, *PDGFR,* and *KIT* in DLBCLs. The percentage of variants that passed the filter with a quality of 100, percentage of variants that did not achieve a quality of 100 (thus not passing the filter), and VAF did not differ significantly among the three histological groups (all *p* > 0.05, Table 4). However, the percentage of variants that passed the filter with a quality of 100 detected using TS decreased in the following order: materials with well-preserved morphology (81.7 ± 15.6) > nuclear streaming samples (80.0 ± 18.9) > necrosis samples (60.8 ± 30.8) (all *p* > 0.05). Furthermore, the percentage of variants that did not pass the filter or achieve a quality of 100 increased in the following order: materials with well-preserved morphology (18.3 ± 15.6) < nuclear streaming samples (20.2 ± 17.9) < necrosis samples (39.2 ± 30.8) (all *p* > 0.05). *TP53* mutations were detected in all 10 cases (materials with well-preserved morphology: 100% (9/9), nuclear streaming samples: 100% (9/9), and necrosis samples: 50% (2/4). The VAFs of *TP53* mutations in all histological cohorts did not differ significantly, being approximately 80% for all cohorts, and occurred in the following order: materials with well-preserved morphology (84.0 ± 22.7) > nuclear streaming samples (83.8 ± 28.7) > necrosis samples (85.8 ± 28.5) (all *p* > 0.05). The high VAFs suggest that the NGS metrics reflect the quality of DNA obtained within the tumor and are not derived from the non-tumor component.

### 3.3. Relationship between Performance in NGS Analysis Using MiSeq and Histological Changes

In the 25 samples, DIN and the percentage of on-target aligned reads were positively correlated (r = 0.671, *p =* 0.003, Figure 3D). All samples except four—one nuclear streaming (DL2018-14) and three necrosis (SC2018-2, DL2018-14, and DL2018-19) samples—showed >80% on-target aligned reads, indicating good alignment with the target region. Although the four samples with <80% on-target aligned reads were all resections, all materials with well-preserved morphology in the same case showed >80% aligned reads, indicating that the results were influenced by the degenerative status in each of the cohorts.

### 3.4. Differences in DIN, RIN, and DV_200_ Based on Sample Volume

As the samples with <80% on-target aligned reads were all resections, we investigated whether the formalin fixed tissue volumes influenced the nucleic acid quality. DIN was significantly higher in the group with small tissue volumes compared to that in the groups with medium and large tissue volumes (both *p* < 0.05, Figure 4A, Appendix A). No significant differences in RIN were detected among the three groups, but the mean RIN in the groups with large, medium, and small sample volumes was 2.05, 2.03, and 1.78, respectively (Figure 4B). DV200 was significantly higher in the group with small tissue volumes compared to that in the groups with medium and large tissue volumes (both *p* < 0.05, Figure 4C). These results indicate that the DNA extracted from smaller tissue samples and RNA extracted from larger tissue samples were better for genetical analysis during 12–24 h of formalin fixation time.

## 4. Discussion

The DNA quality of nuclear streaming samples did not significantly differ from that of materials with well-preserved morphology, whereas necrosis samples were inferior to the other two histological groups. The quality of both DNA and RNA was poor in necrosis samples. The NGS metrics of the nuclear streaming samples did not differ from those of materials with well-preserved morphology, and informative variants were detected. The percentage of on-target aligned reads and variants passing through the filter with high quality was lower in necrosis samples than in the other two sample types. Although some informative variants were detected in the necrosis samples, all failed samples were from the necrosis group, and the quality of DNA was better in smaller samples, such as biopsies.

DIN is a scale used to assess the quality of DNA, and it ranges from 1 (completely degraded) to 10 (completely intact), based on gDNA degradation [7,10,14]. Samples with DIN > 3 are recommended for NGS analysis [7]. The DINs of the samples without degeneration and the nuclear streaming samples were close to 3, similar to those of samples evaluated in the last 4 years, according to the Japanese guidelines (DIN 3.77 ± 1.77) [7,8,10]. The DINs of the necrosis samples were significantly lower. The difference in DIN between nuclear streaming and necrosis samples may be associated with their degenerative processes; necrosis is a biological response that leads to nucleic acid fragmentation [4], whereas nuclear streaming is a histological change caused by sampling stimulation [15,16,17,18]. To enable genetic analysis, particularly for small samples, the nuclear streaming component should be included in the material, without trimming during the pre-analysis evaluation process.

The percentage of on-target aligned reads and variants passing through the filter with high quality was lower in the necrosis samples than in the materials with well-preserved morphology and nuclear streaming samples. Some informative variants were detected in the necrosis samples; however, most samples with an amplicon mean coverage <500× and on-target aligned reads <80% that were considered unsuitable for analysis were necrotic [8]. Because nucleic acid quality most strongly influences NGS quality metrics [1,3,5,6,7], degraded DNA limits the depth of coverage and number of supporting reads, leading to limited variant detection [1,19]. These features are consistent with those of necrosis samples, but not nuclear streaming samples. Only one study has evaluated the usefulness of necrotic tumor samples using TS, reporting that DNA samples obtained from the area of necrosis in colorectal adenocarcinomas are comparable to those of viable tumors [4,20]. However, that was a preliminary finding, and no detailed description of the research results is available [20]. Our results indicate that variants might be identified from necrotic DNA, but these results should be carefully evaluated for false negatives. However, nuclear streaming samples should be considered, as they may contain preserved target DNA and provide genetic information to facilitate therapy.

Furthermore, DIN and DV200 were significantly different regarding formalin fixed tissue sample volumes. Formalin promotes the RNA fragmentation, but inhibits the DNA degradation [8,10]. However, the quality and yield of DNA in large surgical specimens tend to be poorer and less than those of biopsy specimens due to the poor fixation status [10]. Therefore, when using DNA in nuclear streaming samples for genetical analysis, it is preferable to extract it from small tissue sample, such as biopsies.

Although the DNA quality of the degenerative component was low, the VAF and total passing filter (PF) reads of this component did not differ from those of samples with well-preserved morphology. VAF is expressed as the percentage of sequence reads observed by matching a specific mutation derived from the overall coverage of a given locus [21,22]. When used in conjunction with the percentage of cancer cells, VAF can be beneficial in interpreting the mutation type and distribution [7,21,22]. In histological evaluations, non-tumor cells with degenerated features are not counted as tumor content because their DNA is considered incapable of amplification, and it does not enhance the total PF read number. However, our results indicate that not counting degenerated tumor cells will falsely elevate the VAF. Therefore, degenerated non-tumor cells should preferably be assessed when determining the tumor percentage to avoid underestimating the VAF.

Consistent with a previous study, we observed a positive correlation between the percentages of on-target aligned reads and DIN [14]. DIN is a useful indicator of the ability to capture target areas. Particularly, determining DIN is cost-effective and less invasive than re-sampling, as it can be performed during the pre-analysis process. Our results also suggest that evaluating DIN for limited samples with histological degeneration may be necessary to provide more optimal treatment.

Comparative analysis based on RIN and DV_200_ showed that RNA quality in nuclear streaming and necrosis samples was significantly inferior to materials with well-preserved morphology. In NGS analysis, samples with a RIN > 8.0 are of acceptable quality, whereas those with a DV_200_ < 30% are considered unsuitable for analysis [7,8,17]. Herein, the RIN and DV_200_ of the three histological groups did not meet the criteria for NGS, consistent with a previous study conducted in Japan [8] (present RIN, 2.05 ± 0.60; previous RIN, 2.23 ± 0.45). As RNA is less stable than DNA [23], extraction from specimens frozen in liquid nitrogen is recommended [7]; however, this method is not routinely used in Japan. Indicative of poor RNA quality, significantly lower RIN and DV_200_ values were observed for nuclear streaming and necrosis samples; therefore, the use of degenerated samples for RNA-based analyses was challenging. Therefore, DNA-based analysis should be prioritized for samples with histological degeneration, or nucleic acid quality evaluation should be performed before NGS.

This study had some limitations. First, the DNA quality in the samples was low. Furthermore, we were unable to extract all three tissue materials from same patient in all cases, and some samples exceeded the recommended ischemic time of 3 h (Appendix A) [7]. However, a comparison of DIN revealed significant differences among cohorts, and this result is consistent with the NGS metrics of high VAF; the DIN was also positively correlated with the percentages of on-target aligned reads. Therefore, the results can be considered reliable. Second, we did not examine whether RNA extracted from degenerated materials can be used for NGS. Third, we did not make any deep references to germline or somatic loss-of-function mutations in the gene of interest. Future studies should aim to use samples with better nucleic acid quality; obtain all three tissue materials from the same patients in all cases; and employ different library preparations, various genetic analysis modalities, and several variant types.

Based on the results of this study, we recommend pre-analytically evaluating the DIN of nuclear streaming samples and using their DNA for NGS when the sample volume is insufficient. Moreover, if samples with degenerated non-tumor cells, such as lymphocytes, are not completely dissected, wild-type gene reads derived from non-tumor cells could increase the total PF reads and may be misinterpreted as the tumor variant having a lower VAF.

## 5. Conclusions

This is the first study to investigate the use of histologically degenerated samples to determine if these can be used to increase the number of samples available, thus improving diagnosis and treatment. We suggest that DNA extracted from the well-fixed nuclear streaming component can provide useful sequencing data for some target regions, and that degenerated non-tumor cells showing nuclear streaming should be counted when evaluating tumor content to avoid misinterpreting the variant allele frequency.

## Figures and Tables

**Figure 1 cancers-15-01910-f001:**
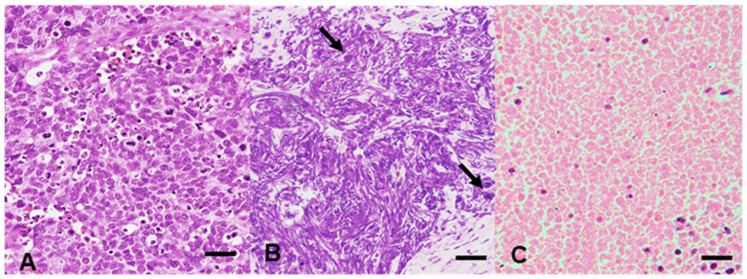
Histological features of the materials with a well-preserved tumor morphological component (**A**), nuclear streaming samples (**B**) (the arrows show the shape-preserved cells), and necrotic samples (**C**). (**A**–**C**) Hematoxylin and eosin staining; original magnification, ×200; bar, 100 μm.

**Figure 2 cancers-15-01910-f002:**
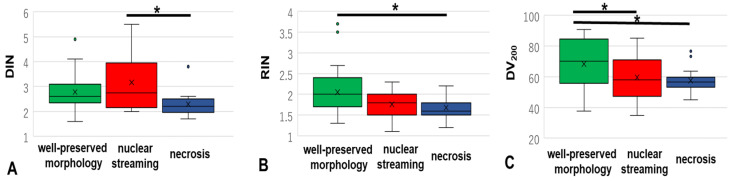
Differences in DIN (**A**), RIN (**B**), and DV_200_ (**C**) among the materials with well-preserved morphology, nuclear streaming, and necrosis groups in SC and DLBCL samples. DIN, DNA integrity number; RIN, RNA integrity number; SC, small cell lung carcinoma; DLBCL, diffuse large-B-cell lymphoma. (*, *p* < 0.05).

**Figure 3 cancers-15-01910-f003:**
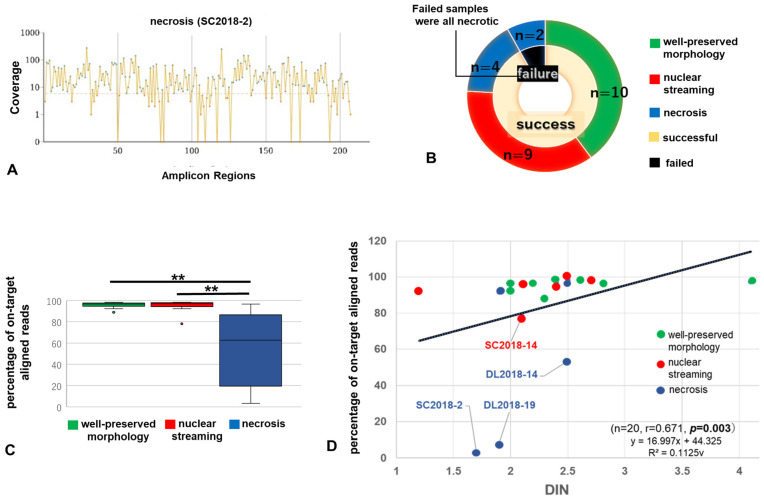
Amplicons with a coverage <×500 (**A**) (necrosis in SC 2018-2). All samples that failed NGS were necrotic (**B**). Differences in percentage of on-target aligned reads among three histological groups (**C**). Positive correlation between the percentage of on-target aligned reads and DIN (**D)** (green, materials with well-preserved tumor morphology; red, nuclear streaming; blue, necrosis). DIN, DNA integrity number; SC, small cell lung carcinoma; DLBCL, diffuse large-B-cell lymphoma. (**, *p* < 0.01).

**Figure 4 cancers-15-01910-f004:**
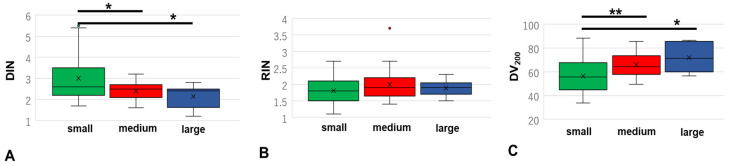
The relationships of DIN (**A**), RIN (**B**), and DV_200_ (**C**) relative to the sample volume: small (≤12 × 12 mm), large (≥24 × 24 mm), and medium (with dimensions other than those of the small and large groups) in SC and DLBCL samples. DIN, DNA integrity number; RIN, RNA integrity number; SC, small cell lung carcinoma; DLBCL, diffuse large-B-cell lymphoma (*, *p* < 0.05; **, *p* < 0.01).

**Table 1 cancers-15-01910-t001:** Comparison of DNA integrity number, RNA integrity number, and DV_200_ between the three histological groups.

	n	Mean	SD	SE	95% CI	vs. Without Degeneration *p*-Value	vs. Nuclear Streaming *p*-Value	vs. Necrosis *p*-Value
**DIN (n = 71)**								
Well-preserved morphology	30	2.78	0.68702	0.12543	2.523–3.037	-	0.222	0.212
Nuclear streaming	28	3.16071	1.11963	0.21159	2.727–3.593	0.222	-	**0.011**
Necrosis	13	2.29230	0.52830	0.14652	1.973–2.612	0.213	**0.011**	
**RIN (n = 72)**								
Well-preserved morphology	31	2.05161	0.59602	0.10704	1.833–2.270	-	**0.05**	**0.026**
Nuclear streaming	30	1.75666	0.31697	0.05787	1.638–1.875	**0.05**	-	0.747
Necrosis	11	1.68181	0.27863	0.08401	1.495–1.869	**0.026**	0.747	-
**DV_200_ (n = 79)**								
Well-preserved morphology	32	68.34093	16.40420	2.89988	62.4266–74.2553	-	**0.041**	**0.031**
Nuclear streaming	30	59.56033	13.70592	2.39534	54.4425–64.6872	**0.041**	-	0.878
Necrosis	17	57.49411	8.15700	1.97836	53.3002–61.6881	0.031	0.878	-

n, number of samples; σ ^2^, non-diversification; SD, standard deviation; SE, standard error; CI, confidence interval; DIN, DNA integrity number; RIN, RNA integrity number; DV_200_, percentage of RNA fragments >200 nt; well-preserved morphology, materials with well-preserved tumor morphology; nuclear streaming, materials with nuclear streaming; necrosis, materials with necrosis; **bold**, statistically significant values.

**Table 2 cancers-15-01910-t002:** Overall sequencing results for the 25 samples obtained from 10 cases.

Year-No.	Status	Organ	Status	Percent Q30 Bases	Total PF Reads	Percentage On-target Aligned Reads	Uniformity of Coverage [Pct > 0.2 × Mean]	Amplicon Mean Coverage	DIN (≥2.3)
SC 2015-1	R	uterus	well-preserved morphology	90.3	690,942	**95.81**	93.24	2803.9	2.0
			nuclear streaming	90.37	396,176	92.33	90.82	1547	1.2
			necrosis	89.64	543,100	71.48	60.87	1494.5	1.4
SC 2015-2	B	lung	well-preserved morphology	79.69	695,098	88.79	67.15	1253.5	2.3
			nuclear streaming	80.92	654,338	97.35	82.61	1358.2	2.5
SC 2018-2	R	lung	well-preserved morphology	80.63	553,474	97.15	88.89	1173.5	-
			nuclear streaming	79.97	482,590	96.97	85.99	1009.7	2.4
			**necrosis**	**86.36**	**270,754**	**3.19**	**78.74**	**29.2**	1.7
SC 2018-3	R	lung	well-preserved morphology	90.33	677,400	96.45	75.36	2737.3	2.8
			necrosis	90.84	668,770	96.52	71.01	2673.8	2.5
SC 2018-4	R	brain	well-preserved morphology	79.74	665,558	95.3	85.99	1347.4	-
			nuclear streaming	79.88	501,748	78.14	82.61	780.7	2.1
SC 2019-8	R	brain	well-preserved morphology	80.94	662,300	97.09	82.13	1402.6	2.2
			nuclear streaming	92.26	396,976	96.18	66.67	1626.7	2.1
SC 2019-9	B	uterus	well-preserved morphology	91.89	949,214	97.39	88.41	3901.8	4.1
			nuclear streaming	81.02	945,082	97.87	93.24	1947.3	-
DL 2018-14	R	intestine	well-preserved morphology	80.36	641,642	98.47	83.57	1375.6	2.6
			nuclear streaming	80.84	648,072	98.14	79.23	1387.9	2.7
			necrosis	76.66	737,332	53.64	82.13	756.4	2.5
DL 2018-19	R	brain	well-preserved morphology	80.46	421,506	92.23	79.23	845.9	2
			nuclear streaming	80.13	593,102	95.93	82.13	1236.5	-
			**necrosis**	**75.26**	**517,606**	**8.05**	**76.33**	**82.6**	1.9
DL 2020-2	R	colon	well-preserved morphology	80.83	464,446	98.34	80.68	1003.7	2.4
			nuclear streaming	80.25	645,404	98.46	87.44	1393.1	2.5
			necrosis	79.49	555,934	91.64	82.61	1079.7	1.9

SC, small cell lung carcinoma; DL, diffuse large B-cell lymphoma; R, resection; B, biopsy; PF, passing filter reads; DIN, DNA integrity number; well-preserved morphology, materials with well-preserved tumor morphology; nuclear streaming, materials with nuclear streaming; necrosis, materials with necrosis; -, unable to measure because the concentration was below the measurement range of the instrument; **bold**, sample with amplicon mean coverage less than ×500; Alt variant freq, variant allele frequency.

**Table 3 cancers-15-01910-t003:** Mutations in 25 samples obtained from 10 cases.

Year-No.	Status	Detected Mutation (n)	Mutations Passed Filter with Quality 100 * (n)	Mutations * Having PP (n)	Gene (Variant)	PP	VAF	Total RD	Recommended RD ^†^
-SC 2015-1	preserved morphology	31	23	3	KDR(T > T/A)	B	66.81	473	9
					TP53(A > A/G)	D	40.4	1222	9
					TP53(G > G/C)	B	24.4	573	42
	nuclear streaming	44	15	2	KDR(T > A/A)	B	98.1	160	2
					TP53(A > A/G)	D	26.9	309	37
	necrosis	28	13	2	KDR(T > T/A)	B	84.8	33	4
					TP53(A > A/G)	D	43	467	17
SC 2015-2	preserved morphology	13	12	1	TP53(G > C/C)	B	100	48	1
	nuclear streaming	14	13	2	TP53(A > C/C)	D	92.7	355	3
					TP53(G > C/C)	B	100	138	1
SC 2018-2	preserved morphology	15	15	4	KIT(G > G/C)	D	47.8	1468	15
					PTEN(A > A/G)	D	52.5	240	13
					KRAS(G >G/T)	D	27.4	880	37
					TP53(G > C/C)	D	90.1	333	3
	nuclear streaming	16	14	4	KIT(G > G/C)	D	47.8	1895	15
					PTEN(A > A/G)	D	50.2	317	14
					PIK3CA(A >A/G)	D	23.1	1617	44
					KRAS(G > G/T)	D	30.5	1189	32
	necrosis	15	2	1	PTEN(A > A/G)	D	75	12	5
SC 2018-3	preserved morphology	30	22	5	ERBB4(C > C/A)	D	35.5	346	27
					KIT(A > A/C)	B	34.5	2041	28
					KDR(T > T/A)	B	63.6	294	10
					MET(A > A/G)	B	41.4	336	18
					TP53(G > C/C)	D	90.1	154	2
	necrosis	26	20	5	ERBB4(C > C/A)	D	76.6	184	5
					KIT(A > A/C)	B	9	1699	195
					KDR(T > T/A)	B	90.1	203	3
					MET(A > A/G)	B	16.4	644	75
					TP53(G > C/C)	D	100	114	1
SC 2018-4	preserved morphology	15	14	2	TP53(T > T/C)	D	68	747	9
					TP53(G > C/C)	D	100	217	1
	nuclear streaming	15	14	2	TP53(T > T/C)	D	91.5	177	3
					TP53(G > C/C)	B	96.9	64	2
SC 2019-8	preserved morphology	10	5	0	-	-	-	-	-
	nuclear streaming	21	17	3	KIT(G > G/C)	B	48.5	1333	15
					TP53(T > T/C)	D	62.7	59	10
					TP53(G > C/C)	B	100	12	1
SC 2019-9	preserved morphology	27	18	2	TP53(G > C/C)	B	99.6	233	2
					STK11(C > C/G)	B	47.4	352	15
	nuclear streaming	16	14	2	TP53(G > C/C)	B	99.5	360	2
					STK11(C > C/G)	B	100	506	1
DL 2018-14	preserved morphology	16	15	2	PDGFR(C > C/G)	B	38.1	1203	25
					TP53(G > C/C)	B	99.8	832	2
	nuclear streaming	17	14	2	PDGFR(C > C/G)	B	34.1	947	28
					TP53(G > C/C)	B	98.8	257	2
	necrosis	16	16	2	PDGFR(C > C/G)	B	32.4	345	30
					TP53(G > C/C)	B	100	55	1
DL 2018-19	preserved morphology	11	9	1	TP53(G > G/C)	B	58.2	79	12
	nuclear streaming	11	9	1	TP53(G > G/C)	B	40.8	172	19
	necrosis	10	5	0	-	-	-	-	-
DL 2020-2	preserved morphology	12	11	2	KIT(A > A/C)	B	45.1	1367	16
					TP53(G > C/C)	B	99.6	634	2
	nuclear streaming	14	11	2	KIT(A > A/C)	B	45.1	2209	16
					TP53(G > C/C)	B	99.8	604	2
	necrosis	14	11	2	KIT(A > A/C)	B	46.7	1537	16
					TP53(G > C/C)	B	99.8	566	2

* Mutations passed filter with quality 100, ^†^ Recommended read depth was calculated using https://github.com/mvasinek/olgen-coverage-limit, accessed on 28 January 2022, where the variant has sufficient corresponding read depth; n, number; PP, PolyPhem; VAF, alt variant number; B, benign; D, probable damage, RD; read depth; -, not detected; preserved morphology, materials with well-preserved tumor morphology; nuclear streaming, materials with nuclear streaming; necrosis, materials with necrosis.

**Table 4 cancers-15-01910-t004:** Comparison of features between materials with well-preserved morphology, with nuclear streaming, and with necrosis.

Metric	n	Mean	SD	SE	95% CI	vs. With Well-Preserved Morphology *p*-Value	vs. Nuclear Streaming *p*-Value	vs. Necrosis *p*-Value
**Percentage of Q30 bases**								
Well-preserved morphology	10	83.517	5.087	1.608	79.8776–87.1564	-	0.954	0.988
Nuclear streaming	9	82.848	4.841	1.613	79.1277–86.5700	0.954	-	0.998
Necrosis	6	83.041	6.771	2.764	75.9355–90.1478	0.988	0.998	-
**Total PF reads**								
Well-preserved morphology	10	642.158	145.218.283	45.922.053	53,8275.10–7406,040.9	-	0.716	0.498
Nuclear streaming	9	584.832	169.773.8	56,591.263	454,332.31–715,331.69	0.716	-	0.910
Necrosis	6	548.916	160.189.302	65,397.009	380,807.64–717,024.36	0.498	0.910	-
**Percentage of on-target aligned reads**								
Well-preserved morphology	10	95.702	3.019	0.954	93.5420–97.8620	-	0.992	**0.001**
Nuclear streaming	9	94.596	6.438	2.146	89.6475–99.5459	0.992	-	**0.02**
Necrosis	6	54.086	40.554	16.556	11.5271–96.6462	**0.001**	**0.02**	-
**Uniformity of coverage**								
Well-preserved morphology	10	82.465	7.491	2.369	77.1056–87.8244	-	0.960	0.237
Nuclear streaming	9	83.415	7.705	2.568	77.4922–89.3389	0.960	-	0.181
Necrosis	6	75.281	8.241	3.364	66.6328–83.9305	0.237	0.181	-
**Variants with filter pass and quality 100 (%)**								
Well-preserved morphology	10	81.70	15.571	4.924	70.56–92.84	-	0.974	0.332
Nuclear streaming	9	80.0	18.000	6.000	66.16–93.84	0.974	-	0.401
Necrosis	6	60.83	30.825	12.684	28.48–93.18	0.332	0.401	-
**Variants without filter pass or with quality < 100 (%)**								
Well-preserved morphology	10	18.30	15.571	4.924	7.16–29.44	-	0.967	0.332
Nuclear streaming	9	20.22	17.894	5.965	6.47–33.98	0.967	-	0.408
Necrosis	6	39.17	30.825	12.584	6.82–71.52	0.332	0.408	-
**Variant allele frequency (*TP53*)**								
Well-preserved morphology	9	84.01	22.695	7.5653	66.566–101.457	-	0.650	0.960
Nuclear streaming	9	83.80	28.657	9.5524	61.772–105.828	0.650	-	0.552.
Necrosis	4	85.77	28.466	14.2334	40.403–130.997	0.960	0.552	-

n, number of samples; PF, passing filter reads; SD, standard deviation; SE, standard error; CI, confidence interval; preserved morphology, materials with well-preserved tumor morphology; nuclear streaming, materials with nuclear streaming; necrosis, materials with necrosis; **bold**, statistically significant values.

## Data Availability

All data were included in manuscript and Appendix A.

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
