# Peer review of "Areas of Crush Nuclear Streaming Should Be Included as Tumor Content in the Era of Molecular Diagnostics"

_cancers, 2023, doi:10.3390/cancers15061910_

Round 1

Reviewer 1 Report

Major comments:

There are many confounding factors regarding the storage conditions for tissue specimens such as DNA/RNA extraction materials. The excellent point of this paper controlling those confounding factors(time to fixation, fixation time, block storage temperature, storage period, etc.) are as comparable as possible other than tissue morphology, the author compared 3 different morphologies within the same tissue samples. However, there is no evidence that the three selected tissue morphologies, 1) without degeneration, 2) nuclear streaming and 3) tumor necrosis, change over time from 1 to 2 and then to 3.

1.    If the author thinks 2 and 3 are causal, please explain them with citation.

2.    If 2 and 3 are unrelated morphological changes, show why you compared them.

The degeneration of tumor cells is a continuous change, not two groups, with degeneration and without degeneration. Differences in the DNA/RNA qualities are expected depending on the degree of necrosis (cases with extensive necrosis vs. cases with only a small area of necrosis). It is expected that the degree of tumor degeneration will differ depending on the tumor mass size and the presence or absence of anticancer drugs, chemotherapy, and radiotherapy.

3.    Is there a difference in DNA quality when dividing the examined cases into large/medium/small tumor volumes?

Minor comments:

Any tissues have proliferating stem-like cells and degenerating/apoptotic cells to maintain normal tissues including tumor tissue, and there is no tissue without degeneration.

This reviewer recommends “well-preserved tumor morphology” to replace “materials without degeneration.”

Reviewer 2 Report

The manuscript by Noda and colleagues reports the evaluation of the quality of DNA/RNA obtained from nuclear streaming and necrosis samples for downstream NGS-based evaluation. My major concern is about the fact that even if DNA/RNA integrity indices are evaluated all the analyses are made just on one DNA application, i.e. targeted DNA sequencing that being based on amplicon capture is usually used for degraded DNA also from paraffin-embedded samples with poor quality. So the novelty seems to be poor. Moreover, the possibility to have interfering signals coming from non-tumor cells is also well known. Accordingly, several methods fort tumor cells isolation have been developed. The paper seems more a technical note that a research article.

Specific points

Simple summary is unclear and should be rewritten.

Abstract: the sentences in the lines 28-30 are in contrast. please clarify.

Line 69: specify that DNA sequencing will be carried out

Patients and samples: the study is informative if you can compare the 3 different samples in the same patients, so the patients for which the 3 samples were not available should not be included in the study.

The assessment of DNA and RNA should be better explained under methods since this is one of the aims of the study.

Differences between the 3 kinds of samples in terms of quality are as expected; however, there is a high intra-group variability as observed by standard deviation. If we consider that the group size is very small, these data should be carefully evaluated. Moreover, we had no data regarding different performances because, as already mentioned, just a targeted amplicon panel was tested on these samples.

Author Response

Respones to Reviewer 2 Comments:

We thank you for your comments and appreciate the time and effort expended to review our manuscript and provide valuable suggestions. We have revised our manuscript accordingly.

  1. Simple summary is unclear and should be rewritten.

Response: We appreciate your useful advice. The Simple Summary has been thoroughly edited and rewritten as follows:

Simple Summary: To expand the pool of samples available for genetic therapy, the quality and utility of DNA and RNA extracted from degenerated tumor tissues were examined. The DNA obtained from the nuclear streaming samples was preserved, enabling the identification of reliable variants using next-generation sequencing (NGS) analysis. The NGS metrics and variant allele frequency of the nuclear streaming samples were not different from those of the control. This suggests that non-tumor cells exhibiting nuclear streaming could be used for genetical analysis, moreover, should be considered as tumor content to avoid misinterpreting the variant allele frequency.

  1. Abstract: the sentences in the lines 28-30 are in contrast. please clarify.

Response: Thank you for your valuable comment. To address your concern, we have changed the sentence from “Necrosis samples were all failed and showed significantly fewer on-target aligned reads and variants. Variant allele frequency and total passing filter reads did not differ among the cohorts.” to ”We revelated DNA in nuclear streaming samples, especially within biopsies could be used for genetical analysis. Moreover, degenerated non-tumor cells should be counted when evaluating tumor content to avoid misinterpreting the variant allele frequency. “

  1. Line 69: specify that DNA sequencing will be carried out

Response: Thank you for your valuable comment. As per your suggestion, we have modified the sentence in Line 96 as follows:

“In this study, we examined the quality of nucleic acids extracted from histologically degenerated tissues, such as samples with necrosis and nuclear streaming, and performed targeted NGS and evaluated sequencing metrics to confirm whether these samples can be used in genetic therapy. ”

  1. Patients and samples: the study is informative if you can compare the 3 different samples in the same patients, so the patients for which the 3 samples were not available should not be included in the study.

 and

  1. Differences between the 3 kinds of samples in terms of quality are as expected; however, there is a high intra-group variability as observed by standard deviation. If we consider that the group size is very small, these data should be carefully evaluated. Moreover, we had no data regarding different performances because, as already mentioned, just a targeted amplicon panel was tested on these samples.

Response: Thank you for your valuable comment. We agree with your point of view. Unfortunately, we were unable to obtain all three types of samples from the same patient in all cases, especially necrosis sample only 11 and necrosis from SC were only four. On the other hand, a comparison of DIN revealed significant differences among cohorts, and NGS results showed high VAF. The DIN was also positively correlated with the percentages of on-target aligned reads. Therefore, we consider the results reliable. However, to address your concern, we have added following portions to the Discussion section (Limitations).

“This study had some limitations. First, the DNA quality in the samples was low. Furthermore, we were unable to extract all three tissue materials from same patient in all cases and some samples exceeded the recommended ischemic time of 3 h (Supplemental Information 4) [7]. However, a comparison of DIN revealed significant differences among cohorts, and this result is consistent with the NGS metrics of high VAF; the DIN was also positively correlated with the percentages of on-target aligned reads.”

“Future studies should aim to use samples with better nucleic acid quality; obtain all three tissue materials from same patients in all cases; and employ different library preparations, various genetic analysis modalities, and several variant types. ”

  1. The assessment of DNA and RNA should be better explained under methods since this is one of the aims of the study. 

Response: Thank you for your valuable comment. As per your suggestion, we have included the following portion in Materials and Methods:

2.2 Assessment of DNA and RNA Quality and Variant Detection Using NGS

DNA and RNA were isolated from each sample, and their purities were measured; their quality was based on RIN, DIN, and DV200. DNA was isolated using a QIAamp DNA formalin-fixed, paraffin-embedded (FFPE) Tissue Kit (Qiagen, Hilden, Germany). RNA was isolated using an miRNeasy FFPE Tissue Kit (Qiagen). The purity of the nucleic acid samples was assessed based on their spectrophotometric absorbance at 260–280 nm (A260/A280) using a NanoDrop 3300 spectrophotometer (Thermo Fisher Scientific, Waltham, MA, USA). RIN, DIN, and DV200 indices and the concentrations of the extracted DNA and RNA samples were quantified using TapeStation 4150 (2015, Agilent Technologies, Santa Clara, CA, USA) based on the genomic DNA (gDNA) and RNA ScreenTape assays. Owing to instrument performance, RIN, DV200, and DIN can provide quality measurements in samples with RNA concentrations > 2.0 ng/μL and gDNA concentrations > 3.0 ng/μL.”

  1. This reviewer recommends “well-preserved tumor morphology” to replace “materials without degeneration.”

Response: Thank you for your valuable comment. As per your suggestion, we have changed the term “materials without degeneration” to “materials with well-preserved tumor morphology” throughout the manuscript.

Reviewer 3 Report

In the introduction, authors can introduce the problem, motivate the problem, and summarize the main contributions

In literature survey, authors can include more similar good papers published in good venues. 

Discuss the limitations of the exsisting works and how the propsed work fills this gap

Discuss the future works

Supplementary Materials: The url given in this sections is not working. Authors are suggested to include the correct link

Authors can make the dataset to the public. It will be helpfull for other researchers

Author Response

Responses to Reviewer 3 Comments:

  1. In the introduction, authors can introduce the problem, motivate the problem, and summarize the main contributions

Response: Thank you for your valuable comment. As per your suggestion, we have added the following portion to the Introduction section.

“In this study, we examined the quality of nucleic acids extracted from histologi-cally degenerated tissues, such as samples with necrosis and nuclear streaming, and performed targeted NGS and evaluated sequencing metrics to confirm whether these samples can be used in genetic therapy. We evaluated their DNA integrity number (DIN), RNA integrity number (RIN), and percentage of RNA fragments with >200 nu-cleotides (DV200), which are commonly used indices to assess nucleic acid quality. For these evaluations, we used materials from small cell lung carcinoma (SC) and diffuse large B-cell lymphoma (DLBCL), in which morphological nuclear degeneration is frequently observed. Our findings can potentially help in increasing the number of samples available for genetic analysis, thereby improving diagnosis and treatment outcomes.”

  1. In literature survey, authors can include more similar good papers published in good venues. 

Response: Thank you for your insightful input. We have cited a reference being available regarding this topic, "Pathology Guidelines on the handling of pathological tissue samples for genomic research," as reference 7.

  1. Kanai, Y.; Nishihara, H.; Miyagi, Y.; Tsuruyama, T.; Taguchi, K.; Katoh, H.; Takeuchi, T.; Gotoh, M.; Kuramoto, J.; Arai, E.; et al. The Japanese Society of Pathology Guidelines on the handling of pathological tissue samples for genomic research: Standard operating procedures based on empirical analyses. Pathol Int. 2018, 68, 63–90. DOI:10.1111/pin.12631.

  1. Discuss the limitations of the exsisting works and how the propsed work fills this gap

   and

  1. Discuss the future works

Response: Thank you for your valuable comment. As per your suggestion, we modified the limitations portion in the Discussion section as follows:

“This study had some limitations. First, the DNA quality in the samples was low. Furthermore, we were unable to extract all three tissue materials from same patient in all cases and some samples exceeded the recommended ischemic time of 3 h (Supple-mental Information 4) [7]. However, a comparison of DIN revealed significant differ-ences among cohorts, and this result is consistent with the NGS metrics of high VAF; the DIN was also positively correlated with the percentages of on-target aligned reads. Therefore, the results can be considered reliable. Second, we did not examine whether RNA extracted from degenerated materials can be used for NGS. Third, we did not make any deep references to germline or somatic loss-of-function mutations in the gene of interest. Future studies should aim to use samples with better nucleic acid quality; obtain all three tissue materials from same patients in all cases; and employ different library preparations, various genetic analysis modalities, and several variant types. ”

  1. Supplementary Materials: The url given in this sections is not working. Authors are suggested to include the correct link. Authors can make the dataset to the public. It will be helpfull for other researchers

Response: We apologize for providing a URL that failed to open. The new URL is as follows:

https://drive.google.com/drive/folders/10tJ1voQbhMpYJs3d1h8l0Jnud5lCznii?usp=sharing

All the data included in this experiment are presented in Supplemental Information.

Round 2

Reviewer 2 Report

The authors has addressed all the previously raised issues. The manuscript has been revised and improved.

It still appears more a technical note than a research article.

Minor points

L11: genetic analysis

Author Response

Response to Reviewer 2 Comments:

We thank you for your comments and We have revised our manuscript accordingly.

Minor points  L11: genetic analysis

Response: We appreciate your useful advice. The term in Simple Summary has been edited and rewritten as follows: To expand the pool of samples available for genetic analysis, the quality and utility of DNA and RNA extracted from degenerated tumor tissues were examined.

Thank you for your valuable comment.